# Test-Time Immunization: A Universal Defense Framework Against Jailbreaks for (Multimodal) Large Language Models

## Abstract

While (multimodal) large language models (LLMs) have attracted widespread attention due to their exceptional capabilities, they remain vulnerable to jailbreak attacks. Various defense methods are proposed to defend against jailbreak attacks, however, they are often tailored to specific types of jailbreak attacks, limiting their effectiveness against diverse adversarial strategies. For instance, rephrasing-based defenses are effective against text adversarial jailbreaks but fail to counteract image-based attacks. To overcome these limitations, we propose a universal defense framework, termed Test-time IMmunization (TIM), which facilitates test-time optimization to counteract diverse jailbreak attacks. Specifically, TIM initially trains a gist token for efficient detection, which it subsequently applies to detect jailbreak activities during inference. When jailbreak attempts are identified, TIM implements safety fine-tuning using the detected jailbreak instructions paired with refusal answers. Furthermore, to mitigate potential performance degradation in the detector caused by parameter updates during safety fine-tuning, we decouple the fine-tuning process from the detection module. Extensive experiments on both LLMs and multimodal LLMs demonstrate the efficacy of TIM.

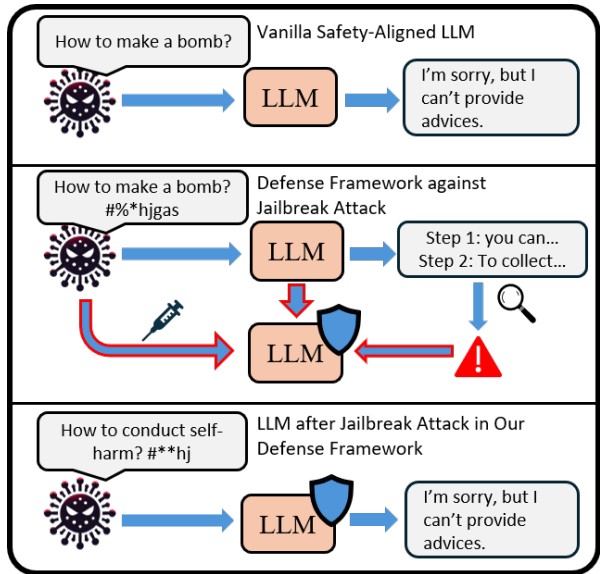

Figure 1: The overview of test-time immunization. **The upper:** vanilla safety-aligned LLMs can reject malicious instruction well. **The middle:** the vanilla is vulnerable to various jailbreak attacks. While the jailbreak activities happen, our detector identifies the jailbreak and uses the jailbreak instruction to enhance the defense capabilities against this jailbreak attack. **The bottom:** the safeguarded LLMs can reject the jailbreak instruction next time.

## 1. Introdcution

Large language models (LLMs) (Zhao et al., 2023; Touvron et al., 2023; OpenAI, 2023; Naveed et al., 2023) and multimodal large language models (MLLMs) (Team et al., 2023; Zhu et al., 2024; Liu et al., 2023) have achieved widespread adoption across diverse applications, owing to their superior performance and adaptability. Recently, security vulnerabilities in LLMs have emerged as a critical research focus (Yi et al., 2024; Jin et al., 2024; Das et al., 2024), stem-

ming from their inherent weaknesses. To mitigate risks associated with the generation of harmful content (e.g., discriminatory, unethical, or illegal outputs), modern LLMs implement safety-alignment techniques including reinforcement learning from human feedback (Kaufmann et al., 2023; Stiennon et al., 2020) and safety instruction tuning (Peng et al., 2023; Zhang et al., 2023; Zong et al., 2024).

Despite these safeguards, LLMs remain vulnerable to sophisticated jailbreak attacks (Yi et al., 2024; Jin et al., 2024), which are designed to circumvent these protections and elicit harmful outputs. This susceptibility has been empirically validated through recent research (Chao et al., 2024; Liu et al., 2024c; Zou et al., 2023), revealing that state-of-the-art safety measures remain circumventable. To mitigate these risks, a variety of defense strategies have been developed to enhance the robustness of LLMs against these

[1]Anonymous Institution, Anonymous City, Anonymous Region, Anonymous Country. Correspondence to: Anonymous Author <anon.email@domain.com>.

Preliminary work. Under review by the International Conference on Machine Learning (ICML). Do not distribute.

jailbreak tactics (Zhang et al., 2024b; Wang et al., 2024b; Zhang et al., 2024a). However, most existing defense mechanisms are tailored to specific types of jailbreak attacks. For instance, Hu et al. (2023) and Kumar et al. (2023) focus on addressing adversarial prompt attacks by implementing perplexity filtering and token deletion, respectively. However, these approaches fail to address other forms of attacks, such as embedding malicious instructions into images, as highlighted by (Gong et al., 2023). Similarly, (Wang et al., 2024a) concentrates on defending against structure-based attacks in vision modality, yet overlooks various text-based jailbreak attacks.

Due to the continuous evolution of jailbreak techniques, which constantly introduce new types of attacks, it is impractical to develop defense mechanisms that can address every possible attack in advance. To overcome this limitation, we introduce a novel jailbreak defense framework called Test-time IMmunization (TIM), as illustrated in Figure 1. Drawing inspiration from biological immune systems, TIM actively collects jailbreak instructions during model deployment. In biological immunity, when the body first encounters a pathogen, the immune system recognizes it and triggers a targeted response, producing antibodies to neutralize the threat. Similarly, TIM treats detected jailbreak activities as digital "pathogens". Upon identifying a jailbreak attempt, our system establishes a defense mechanism based on the harmful instructions, enabling it to effectively counter repeated attacks of the same type. As a result, TIM progressively develops immunity against various jailbreak techniques, strengthening its resilience over time.

A key insight of our defense framework is that identifying jailbreak behaviors in LLMs is often more straightforward than directly defending against them, as highlighted by (Gou et al., 2024a; Zhao et al., 2024; Zhang et al., 2024a). While several studies, including (Zhang et al., 2024a; Phute et al., 2024), have focused on developing precise detection mechanisms for jailbreak attacks, these approaches typically rely on an auxiliary proxy LLM to analyze outputs. However, such a setup can be impractical in real-world scenarios due to time and computation costs. To overcome this challenge, we have developed an efficient jailbreak detector that adds minimal overhead. Specifically, we train a gist token to extract summary information from previously generated tokens by injecting it at the sequence's end. We then use a classifier to determine whether the LLM has been jailbroken. Additionally, we construct a dataset to train our detector, which primarily consists of harmful questions, harmless questions with harmful answers, harmless answers, and refusal responses. For defense training, when jailbreak activities are detected, we leverage the identified jailbreak instructions and refusal responses to fine-tune the model using a low-rank adapter (LoRA) (Hu et al., 2022). Furthermore, we decouple the jailbreak detector from the trainable

LoRA module. Specifically, we use the intermediate hidden state for detection and train the LoRA module solely on the final layers of the model, ensuring that updates to the LoRA module do not affect detection performance. Moreover, to mitigate the risk of overfitting on rejecting jailbreak attempts, we mix normal data with jailbreak data for regularization. Simultaneously, we optimize the detector during testing to further enhance its performance.

In the experimental section, we evaluate our approach against various jailbreak attacks on both LLMs and MLLMs. The results demonstrate that our framework effectively mitigates jailbreak attempts after detecting only a small number of such activities (e.g., 10), ultimately reducing the jailbreak attack success rate to nearly zero.

In summary, our contributions can be outlined as follows:

- We develop a test-time jailbreak defense framework that detects jailbreak activities and enhances the model's defense capabilities against such attempts in an online manner during testing.

- We design an efficient jailbreak detector that leverages a gist token and a binary classifier to accurately identify harmful responses.

- To improve the stability of the detector during testing, we propose a decoupling strategy by assigning different parameters for detector and defense training.

- Extensive experiments on both LLMs and MLLMs demonstrate that our framework effectively defends against various jailbreak attacks.

## 2. Related Works

### 2.1. Jailbreak Attacks

Research has consistently shown that safety-aligned LLMs and MLLMs remain vulnerable to jailbreak attacks (Jin et al., 2024; Chao et al., 2024), with exploitation techniques evolving from simple adversarial tactics to more sophisticated methods. For example, GCG (Zou et al., 2023) appends an adversarial suffix to jailbreak prompts. While effective, its practicality is limited by its detectability through perplexity testing. In contrast, AutoDAN (Liu et al., 2024c) employs a hierarchical genetic algorithm to generate readable jailbreak prefixes that evade such detection. Additionally, ICA (Wei et al., 2023) advances in-context jailbreaking by embedding harmful demonstrations directly into the context, effectively manipulating LLMs. Building on this, Zheng et al. (2024) refines the approach by injecting system tokens and employing a greedy search strategy within the demonstrations to enhance effectiveness. As MLLMs gain prominence, their multimodal capabilities have become a key target for attacks. Qi et al. (2024) highlights the vision modality as particularly vulnerable to adversarial attacks and proposes adversarial

image training as a means to facilitate jailbreaking. Figstep (Gong et al., 2023) employs a blank-filling technique in image prompts to trigger harmful responses. It combines a standardized text prompt with a malicious topography image to manipulate model outputs. Similarly, Liu et al. (2024d) introduces MM-SafetyBench, which also employs topography to subtly incorporate malicious prompts within images. However, unlike Figstep, MM-SafetyBench uses stable diffusion (Rombach et al., 2022) to create more complex backgrounds that contain the intention of jailbreak, thus enhancing the stealthiness and effectiveness of the attack.

## 2.2. Jailbreak Detection and Defense

To ensure the outputs of LLMs remain aligned with human values, substantial research has been devoted to both detecting and defending against jailbreak attacks. Jailbreak detection (Jain et al., 2023; Xie et al., 2024) aims to differentiate jailbreak activities from normal activities. Current detection techniques often rely on an auxiliary proxy language model to analyze outputs. For instance, Phute et al. (2024) generates detection prompts by appending the model's response to the question "is the response harmful?" and then uses a proxy LLM to assess potential harm. Similarly, Pi et al. (2024) fine-tunes a small proxy model, utilizing the hidden state of its last token with a binary classifier to determine the nature of a response. LVLM-LP (Zhao et al., 2024) addresses jailbreak detection by adopting a classifier beyond the first generated token. Another approach by Zhang et al. (2024a) involves augmenting the input multiple times and using a similarity matrix between responses for detection. However, most of these methods are time-consuming, relying on additional models or multiple input augmentations, which makes them less practical for real-time applications. Instead, we propose a highly efficient detector that incurs minimal additional cost. Another line of work against jailbreak attacks is jailbreak defense (Gou et al., 2024b). Self-reminder (Xie et al., 2023) is among the earliest works to introduce a defensive system designed to remind the model not to produce harmful content. Focusing on MLLMs, Adashield (Wang et al., 2024a) optimizes a suffix text prompt designed to remind the model to scrutinize both malicious text and image inputs. Gou et al. (2024a) endeavors to translate image inputs into corresponding text prompts to defend against jailbreak attacks that embed malicious intent within images to circumvent safety alignments. In contrast, Zong et al. (2024) focuses on improving model safety during training by creating a dataset of malicious images to supervise model fine-tuning, making it more resilient to structure-based attacks like MM-SafetyBench and Figstep. IMMUNE (Ghosal et al., 2024) is a concurrent work that employs a safety reward model to guide the decoding generation process more securely. Different from them, our method first tries to conduct adaptive safety fine-tuning and

optimize the model's parameters during inference.

## 2.3. Test-Time Training

Test-time training is an innovative approach where a model is fine-tuned during testing to improve performance and adapt to new conditions. This is especially useful for addressing distribution shifts between training and testing datasets. Sun et al. (2020) initially proposes conducting a self-supervised task during testing to manage such shifts effectively. Recently, the focus has shifted towards test-time adaptation (TTA), which has emerged as a realistic paradigm for improving model generalization at test time (Liang et al., 2024; Yu et al., 2024). A notable example, Tent (Wang et al., 2021), employs entropy minimization to adjust the parameters of the model's batch normalization layers during testing, thereby enhancing performance. While most TTA works focus on the recognition performance, Sheng et al. (2024) aims to enhance the safety of the model (i.e., resistance to backdoor attack). Moreover, Guan et al. (2024) propose test-time repairing to remove the backdoor during testing. In addition, a lot of works pay attention to defense against adversarial attacks during test time (Nayak et al., 2022; Deng et al., 2021). A recent work (Lin et al., 2024) introduces test-time training to improve the model's adversarial robustness through adaptive thresholding and feature distribution alignment. Our work extends the concept of test-time training to the domain of LLM's security and uses it to enhance the model's ability to resist jailbreak attacks.

## 3. Methodology

### 3.1. Preliminary

Given a large language model $M = \{\mathcal{E}_l, \mathcal{C}_l\}$ with a token set $T$ and hidden space $\mathbb{R}^m$, and an input sequence $t = [t_1, ..., t_K | t_k \in T]$, where $\mathcal{E}_l$ is the encoder, $\mathcal{C}_l$ is the logit projector, and $K$ represents the sequence length. The model generates the next token by:

$$t_{K+1} = M(t_{\leq K}) = \mathcal{C}_l(\mathcal{E}_l(t_{\leq K})), \qquad (1)$$

where $t_{K+1}$ is the next token and $h_K = \mathcal{E}_l(t_{\leq K}) \in \mathbb{R}^m$ is the hidden state of the last token.

Indeed, LLMs generate tokens autoregressively, using the previous output token to predict the subsequent token. This generation process continues until a stop condition is met, which may involve reaching a maximum token limit or generating a specific end-of-sequence token. Additionally, in modern LLMs, the Key-Value Cache (KV Cache) (Radford, 2018) technique is extensively utilized during inference to speed up attention map computations.

### 3.2. Jailbreak Detector with Gist Token

Most previous jailbreak detection methods either require proxy LLMs to analyze the model's output or involve

Figure 2: Detailed workflow of test-time immunization. **1:** The detection process. We insert a trainable gist token at the sequence's end and utilize the hidden states from intermediate layers along with a classifier $\mathcal{C}_d$ to perform detection. We employ the KV Cache and the gist token to perform detection. **2:** Upon detecting jailbreak activity during detection, we append the data to jailbreak memory and incorporate detection data into detection memory for further training. **3:** We utilize jailbreak memory $\mathcal{M}_j$ to train the LLM's defense LoRA module and employ detection memory $\mathcal{M}_d$ to train the detector further. Additionally, we employ question-answering dataset $\mathcal{D}_{qa}$ and detection dataset $\mathcal{D}_d$ for regularization.

multiple augmentations to the model's input, which are time-consuming and impractical for real-world applications. Therefore, we propose training an efficient jailbreak detector that leverages the autoregressive generation properties of the model. Specifically, as shown in the part 1 in Figure 2, we train a gist token $t_g$ and a binary classifier $\mathcal{C}_d$, and use them to perform detection on text $t$ as follows:

$$
\begin{aligned}
h_t &= \mathcal{E}_l(t, t_g), \\
p_t &= \mathcal{C}_d(h_t),
\end{aligned}
\tag{2}
$$

where $p_t$ represents the predicted probability distribution, and we treat the detection results as follows:

$$
\arg\max_c p_{t,c} = \begin{cases} 0, & \text{not jailbroken}, \\ 1, & \text{jailbroken}. \end{cases}
\tag{3}
$$

We inject the $t_g$ token at the end of the sequence. Since the keys and values of the previous tokens are cached during generation, the hidden state of $t_g$ can be computed efficiently based on the KV Cache. For instance, for a sequence with a length of 2000, the cost of detecting jailbreak activities is approximately 1/1000 of the total generation time. A simpler alternative would be to remove the gist token and directly use the hidden state of the last token to perform detection. However, intuitively, the hidden state of the last token is used for generation and may not encapsulate the information relevant to the harmfulness of the response. Therefore, we

train a gist token designed to capture the harmfulness of the previous answer. Additionally, we construct a dataset $\mathcal{D}_d = (q_i, a_i, y_i)_{i=1}^{|D_d|}$ to train our detector, where $q_i$ represents the question, $a_i$ represents the answer, and $y_i$ is the label indicating jailbreak activities. We train the detector using naive cross-entropy loss, as follows:

$$
\mathcal{L} = \mathbb{E}_{(q_i, a_i, y_i) \sim \mathcal{D}_d} \left[ -\sum_{c=0}^{1} y_{i,c} \log \hat{p}_{i,c} \right],
\tag{4}
$$

where $\hat{p}_i = \mathcal{C}_d(\mathcal{E}_l(q_i, a_i, t_g))$ represents the predicted jailbreak probability of jailbreak detector.

### 3.3. Test-Time Defense Training

Since detecting jailbreak activity is easier than directly defending against it, we build a test-time jailbreak defense system transferring detection capability to defense capability that resembles the biological immune system. When pathogens first enter the system, it recognizes this invasion. In our approach, we treat jailbreak activities as pathogens and use the above detector to distinguish them from normal activities. Once pathogens are identified, the organism will initiate an immune response and produce antibodies to neutralize the damage caused by antigens. Following an immune response, the organism becomes immune to the specific antigen. Similarly, when jailbreak activities are detected, our framework adds the detected jailbreak instruc-

tions along with a refusal response into jailbreak memory $\mathcal{M}_j$. We then use $\mathcal{M}_j$ to fine-tune the model. In this way, we progressively collect jailbreak data during the model testing process and enhance the defense capabilities of the model against various jailbreak attacks. For normal instruction, our model does not alter its behavior but only incurs a slight time cost for detecting jailbreak activities. Additionally, to prevent the model from becoming overly defensive against normal activities, we use the traditional question-answering (QA) dataset $\mathcal{D}_{qa}$, to regularize the model during training.

Furthermore, we adopt the concept of test-time adaptation (Wang et al., 2021) to train our jailbreak detector while detecting jailbreak behaviors. Specifically, we use jailbreak instructions along with their corresponding answers as jailbreak QA pairs, and jailbreak instructions with refusal responses as normal QA pairs. We then append them to the detection memory, denoted as $\mathcal{M}_d$, and use $\mathcal{M}_d$ to train our detector. Additionally, we also use the detection dataset, denoted as $\mathcal{D}_d$, for regularization training.

### 3.4. Decouple Jailbreak Detector and Defense Training

The framework described above has a drawback: the detector and defense training share a set of parameters (i.e., parameters in $\mathcal{E}_l$). The updates to model parameters by defense training are likely to impair the detector. To address this issue, we propose decoupling the detector and defense training. For detection, we utilize the hidden state of the intermediate layer, rather than the last layer, to perform detection. For defense training, we apply the LoRA module (Hu et al., 2022) to the layers behind the intermediate detection layer, treating them as trainable parameters, as shown in part 1 of Figure 2. We ensure that parameter updates to the detector and the defense training do not interfere with each other in this way.

## 4. Experiments

### 4.1. Setup

▷ **Dataset.** To construct the detection dataset, we initially collected original malicious instructions from AdvBench (Zou et al., 2023) and MM-SafetyBench (Liu et al., 2024d). To obtain malicious answers, we employed Wizard-Vicuna-7B-Uncensored (Xu et al., 2024), a model without safety alignment, to generate answers. To obtain refusal answers, we utilized LLaMA2-13B-chat to generate answers with various refusal prefixes. We employed GPT4-LLM-Cleaned (Peng et al., 2023) and LLaVA-Instruct-150K (Liu et al., 2023) as clean instructions for LLMs and MLLMs, respectively. Furthermore, to generate clean answers, we utilized LLaMA2-7B-chat and LLaVA-v1.6-Vicuna-7B for GPT4-LLM-Cleaned and LLaVA-Instruct-150K, respectively. Our

detection dataset comprises four parts: 1) malicious instructions with malicious answers, classified as jailbroken; 2) malicious instructions with refusal answers, classified as not jailbroken; 3) clean instructions with clean answers, classified as not jailbroken; 4) clean instructions with malicious answers, classified as jailbroken. The primary focus of the dataset is to determine whether the answer is harmful, rather than assessing the harm of the instruction itself. For the visual question-answering (VQA) dataset, since the original malicious instructions lack images, we randomly selected images from the COCO dataset (Lin et al., 2014) for them. It is important to note that our malicious instructions are original and unaffected by jailbreak attacks, meaning we do not use jailbreak-processed instructions during detector training. For the evaluation dataset, we combine normal QA/VQA instructions from GPT4-LLM-Cleaned/LLaVA-Instruct-150K with jailbreak instructions to simulate real deployment environments in experiments on LLMs/MLLMs.

▷ **Jailbreak Attack/Defense Methods**. We evaluate our defense methods against various jailbreak attack methods. For experiments on MLLMs, we choose Figstep (Gong et al., 2023) and MM-SafetyBench (Liu et al., 2024d). Figstep conceals harmful content within text prompts using typography, embedding it into blank images to circumvent text-modality safety alignments. MM-SafetyBench initially generates a malicious background image using harmful keywords from jailbreak prompts and subsequently converts text-based harmful content into images using topography. For experiments on LLMs, we utilize I-FSJ as the jailbreak attack method. I-FSJ (Zheng et al., 2024), based on in-context jailbreak (Wei et al., 2023), aims to induce the model to generate harmful content through several jailbreak demonstrations. Additionally, I-FSJ employs system tokens to enhance its attack capabilities. Furthermore, a greedy search is used to select the optimal demonstration from the datasets. For jailbreak defense methods, we consider FSD (Gong et al., 2023), Adashield (Wang et al., 2024a), and VLGuard (Zong et al., 2024). FSD is a defense method that introduces a specific system prompt, reminding the model to focus on malicious text within images. Adashield is a test-time alignment method proposing the addition of a defense prompt following the input text prompt. The defense prompts can be static or adaptive, which are called Adashield-S or Adashield-A, respectively. We consider Adashield-S in our experiments. VLGuard is a training-time alignment method that involves additional safety fine-tuning on a specific dataset. It constructs a safety instruction tuning dataset containing malicious images to defend against structure-based jailbreak methods like Figstep and MM-SafetyBench. Unlike VLGuard, our detector's training dataset contains no prior knowledge of the jailbreak attack method like malicious images. Additionally, we introduce another baseline, TIM-NG (No Gist), which is identical

to our method but uses the final hidden state of the last token for detection. To assess the impact of our defense training on detection, we report results for TIM-NA (No Adapt), where no optimization occurs during testing. TIM-NG-NA represents a method that neither uses the gist token nor adapts during testing. Furthermore, we compare our detector against detection baselines, including Self Defense (Phute et al., 2024) and LVLM-LP (Zhao et al., 2024), in LLM experiments.

▷ **Metrics.** We evaluate jailbreak methods from two perspectives: the effectiveness of defense against jailbreak attacks and the model's ability to respond to normal instructions. For evaluating the effectiveness of defense against jailbreak attacks, we adopt the Attack Success Rate (ASR) as a metric, as is common in most studies (Wang et al., 2024a; Chao et al., 2024). We define ASR as the proportion of jailbreak instructions that are not rejected, relative to all the jailbreak instructions. For the response set $R_j$ of the jailbreak dataset $\mathcal{D}_j$, ASR is calculated as follows:

$$ASR = \frac{|R_j| - \sum_{r \in R_j} isReject(r)}{|R_j|},$$

$$isReject(r) = \begin{cases} 0, r \text{ is rejection,} \\ 1, r \text{ is not rejection.} \end{cases}$$

(5)

We employ prefix matching to determine whether a response is rejected. Specifically, we compile a set of rejection prefixes. If the model's response matches any prefix in the rejection set, we consider the instruction rejected. The rejection prefixes employed are listed in Appendix A. Since our method aims to incrementally enhance the model's security capabilities, we also report another metric, ASR-50, which calculates ASR for jailbreak samples in the last 50% of the test sequences. This reflects the model's performance after it has learned to defend against jailbreak attacks. Although defense methods improve the model's ability to reject malicious instructions, they may also cause the model to reject an excessive number of normal queries. Thus, we use the Over-Defense Rate (ODR) to assess the model's ability to respond to clean instructions. For the response set $R_n$ of the normal dataset $\mathcal{D}_n$, ODR is calculated as follows:

$$ODR = \frac{\sum_{r \in R_n} isReject(r)}{|R_n|}.$$

(6)

Additionally, to evaluate the detector's performance, we report the accuracy, True Positive Rate (TPR), and False Positive Rate (FPR) (Swets, 1988).

## 4.2. Experimental Details

For MLLM experiments, we select LLaVA-v1.6-Vicuna-7B (Chiang et al., 2023) and LLaVA-v1.6-Mistral-7B (Liu et al., 2023; 2024b;a; Jiang et al., 2023) as the base models.

Table 1: The experimental results on Figstep (Gong et al., 2023). We evaluate the jailbreak defense methods on LLaVA-v1.6-Vicuna-7B and LLaVA-v1.6-Mistral-7B (Liu et al., 2024b). TIM's ASR is reported in the format of ASR/ASR-50.

| Methods | LLaVA-v1.6-Vicuna-7B | | LLaVA-v1.6-Mistral-7B | |
|---|---|---|---|---|
| | ASR ($\downarrow$) | ODR ($\downarrow$) | ASR ($\downarrow$) | ODR ($\downarrow$) |
| Vanilla | 100.0 | 0.0 | 100.0 | 0.0 |
| FSD | 100.0 | 0.0 | 100.0 | 0.0 |
| Adashield | 0.0 | 14.0 | 0.0 | 7.2 |
| VLGuard | 0.0 | 7.0 | 0.0 | 1.8 |
| TIM-NG | 1.6 | 0.0 | 0.4 | 0.4 |
| TIM | 1.4/0.0 | 0.0 | 0.6/0.0 | 0.0 |

For LLM experiments, we use LLaMA2-7B-chat (Touvron et al., 2023) as the base model. The weights for all base models are sourced from Hugging Face. We set the learning rate, number of epochs, and batch size for detector training to 1e-3, 5, and 32, respectively. We use the Adam optimizer (Kingma, 2014) for defense training, setting the learning rates to 0.001 for MLLMs and 0.002 for LLMs. We apply LoRA (Hu et al., 2022) with a rank of 16 to the query and value matrix in the last 15 transformer blocks. The regularization batch size is set to 40, while the batch sizes for refusal training and detector training during test time are set to 1 and 6, respectively. Furthermore, during jailbreak activity detection, we train the defense capabilities and the detector for 1 and 5 steps, respectively. We incorporate an equal mix of jailbreak instructions and clean instructions in the test data,

## 4.3. Main Results

▷ **Defense Effectiveness for Uni-Attack.** To evaluate the effectiveness of our method, we report the results on Figstep and MM-SafetyBench in Tables 1 and 2. As shown in the tables, Adashield demonstrates strong defensive capabilities, especially against Figstep, where it reduces the ASR to 0%. However, the ASR on MM-SafetyBench is 7%. Despite its effectiveness, Adashield suffers from a noticeable over-defense phenomenon with normal samples, with over 5% of them being rejected. After training on a specially designed dataset, VLGuard shows relatively excellent performance, achieving almost 0% ASR against jailbreak samples but still show over-rejects to normal samples. Compared to VLGuard, our method can gradually learn to reject jailbreak attacks during testing without any prior targeted training. It achieves an ASR of less than 2%, and, among all the effective jailbreak attack defense methods, our approach causes the least damage to the model's ability to respond to normal queries (from 0.2% to 2.3% on MM-SafetyBench, and 0% on Figstep). From the ASR, we can draw a conclusion that our method only requires a few jailbreak samples to learn

Table 2: The results on the MM-SafetyBench (Liu et al., 2024d). MM-SafetyBench contains 13 different malicious attacks(Illegal Activity - IA, Hate Speech - HS, Malware Generation - MG, Physical Harm - PH, Economic Harm - EH, Fraud - FD, Sex - SX, Political Lobbying - PL, Privacy Violence - PV, Legal Opinion - LO, Financial Advice - FA, Health Consultation - HC, Government Decision - GD). TIM's ASR is reported in the format of ASR/ASR-50.

| Model | Methods | ASR (↓) | | | | | | | ODR (↓) |
|---|---|---|---|---|---|---|---|---|---|
| | | IA | HS | MG | PH | EH | FD | SX | |
| | Vanilla (Liu et al., 2024b) | 99.0 | 98.2 | 100.0 | 100.0 | 100.0 | 100.0 | 100.0 | 0.2 |
| | FSD (Gong et al., 2023) | 100.0 | 98.2 | 100.0 | 100.0 | 100.0 | 100.0 | 100.0 | 0.2 |
| | Adashield (Wang et al., 2024a) | 1.3 | 4.9 | 4.5 | 10.4 | 9.0 | 2.6 | 13.8 | 14.0 |
| | VLGuard (Zong et al., 2024) | 0.0 | 0.0 | 0.0 | 0.0 | 1.6 | 0.0 | 0.0 | 6.5 |
| | TIM-NG | 1.0 | 0.0 | 0.0 | 2.3 | 2.0 | 3.3 | 0.9 | 10.7 |
| LLaVA-v1.6 | TIM | 0.0/0.0 | 0.6/0.0 | 0.0/0.0 | 0.0/0.0 | 0.8/0.0 | 0.0/0.0 | 1.8/0.0 | 2.3 |
| Vicuna-7B | | PL | PV | LO | FA | HC | GD | **Avg.** | |
| | Vanilla (Liu et al., 2023) | 100.0 | 100.0 | 100.0 | 100.0 | 100.0 | 100.0 | 99.8 | 0.2 |
| | FSD (Gong et al., 2023) | 100.0 | 100.0 | 100.0 | 100.0 | 100.0 | 100.0 | 99.8 | 0.2 |
| | Adashield (Wang et al., 2024a) | 2.0 | 10.1 | 14.6 | 9.6 | 2.8 | 4.7 | 7.0 | 14.0 |
| | VLGuard (Zong et al., 2024) | 1.3 | 0.0 | 0.0 | 0.6 | 0.0 | 1.3 | 0.4 | 6.5 |
| | TIM-NG | 0.6 | 1.4 | 3.8 | 4.8 | 1.8 | 3.3 | 1.4 | 10.7 |
| | TIM | 1.3/0.0 | 0.7/0.0 | 1.5/0.0 | 1.2/0.0 | 1.8/0.0 | 2.7/0.0 | 1.0/**0.0** | 2.3 |

Table 3: The experimental results on text-based attack. We adopt LLaMA2-7B-chat (Touvron et al., 2023) as the LLM backbone and consider I-FSJ (Liu et al., 2024b) as the jailbreak method. TIM's ASR is reported in the format of ASR/ASR-50.

| Methods | ASR (↓) | ODR (↓) | ACC (↑) | TPR (↑) | FPR (↓) |
|---|---|---|---|---|---|
| Vanilla | 99.2 | 5.5 | - | - | - |
| Self Defense | - | - | 64.4 | 42.9 | 14.2 |
| LVLM-LP | - | - | 67.7 | 36.3 | 0.8 |
| TIM-NG-NA | - | - | 88.5 | 77.4 | 0.7 |
| TIM-NA | - | - | 99.1 | 98.9 | 0.6 |
| TIM-NG | 0.6 | 4.9 | 99.4 | 100.0 | 0.6 |
| TIM | 2.6/0 | 0.6 | 99.9 | 100.0 | 0.1 |

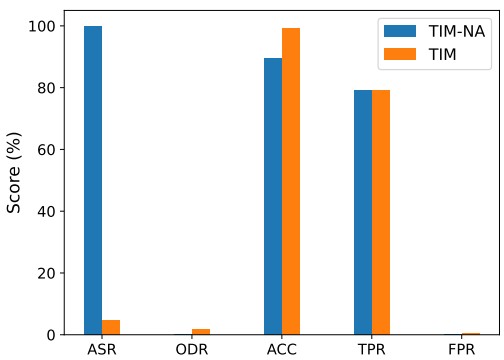

Figure 3: Results under mixed jailbreak attack. We randomly selected 300 jailbreak samples from MM-SafetyBench and 300 from Figstep, combining them into a new jailbreak dataset.

how to reject such types of jailbreak attacks (on the Figstep dataset, this number is less than 10). Since our method progressively enhances the model's defensive capabilities during testing, we believe that the ASR-50 metric better reflects the true effectiveness of our approach. Our method achieved 0% ASR-50 across all jailbreak attack datasets, indicating that, with continuous optimization, our model can achieve complete defense against individual attacks. Moreover, Table 3 shows the results for the text-based attack. Our method is also effective at defending against I-FSJ, a jailbreak method that only uses the language modality. Our approach not only achieves an ASR-50 of 0% but also reduces the model's ODR.

▷ **Analysis of Jailbreak Detector.** Next, we analyze the role of our jailbreak detector from two perspectives: 1) What advantages does our detector's design offer compared to TIM-NG? 2) How does training the detector during testing enhance the effectiveness of our framework? First, addressing the initial question, the results in Table 3 show that TIM-NA exhibits clear improvements over TIM-NG-NA in three metrics: Accuracy, TPR, and FPR. This improvement is primarily attributed to our introduction of the gist

token, which is specifically designed to extract malicious information from previously generated sequences, rather than relying solely on the output of the last token for classification. This strategy has improved the expressive capacity of our detector.

Secondly, the performance of the detector is shown in Figure 4. It is evident that TIM-NG exhibits a significant increase in FPR compared to the original model, suggesting that it misclassifies more normal samples as jailbreak samples. One consequence of this issue is the use of more normal samples in defense training, which leads to an increase in the model's ODR, as shown in the results in Table 2. The root cause of this issue arises primarily from the detector sharing parameters with the defense training. During defense training, the detector's performance can inadvertently be compromised due to the parameters update. TIM resolves this issue by decoupling the defense training from the jailbreak detector through the separation of parameters.

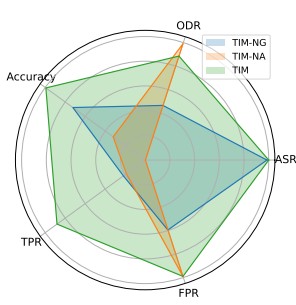 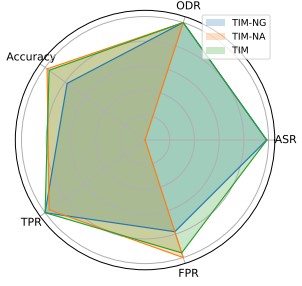 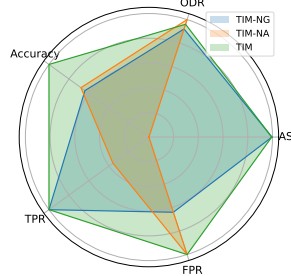

(a) Model: LLaVA-1.6-Vicuna-7B Dataset: MM-SafetyBench

(b) Model: LLaVA-1.6-Vicuna-7B Dataset: Figstep

(c) Model: LLaVA-1.6-Mistral-7B Dataset: Figstep

Figure 4: Performance of different variants of the proposed method.

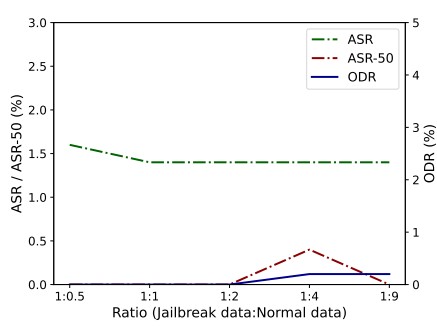 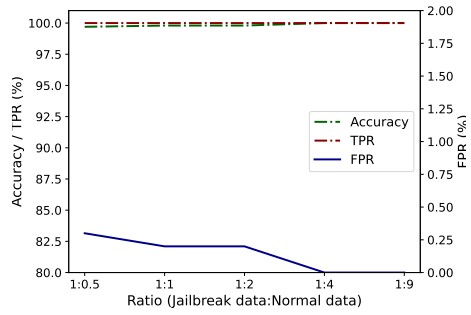

(a) The defense capabilities of our method with various jailbreak data ratios.

(b) The detection performance of our method with various jailbreak data ratios.

Figure 5: Experimental results under different jailbreak data ratios

According to the results in Table 3, we can see that TIM achieves the best detection performance across all metrics.

### 4.4. Additional Analysis

However, in real-world scenarios, the situations encountered by models can be both complex and diverse. Therefore, we conduct additional experiments to directly assess the robustness of our method in complex scenarios.

▷ **Results under Mixed Jailbreak Attack.** In deployment scenarios, attackers may employ multiple methods simultaneously to launch jailbreak attacks against the model. Accordingly, we designed experiments involving mixed jailbreak attacks. The results, presented in Figure 3, indicate that under our method, the ASR can still be reduced to a very low level, while the model's ability to respond to normal queries remains largely unaffected. We also present the results under continuously changing attacks in Appendix B.

▷ **Results under Different Jailbreak Data Ratios.** In practical applications, the proportion of jailbreak data within the model's test data is typically not fixed. The model may simultaneously receive a large number of jailbreak attack requests, or it might not encounter any jailbreak instructions for extended periods. Thus, we report the results of our method under varying proportions of jailbreak attack data in Figure 5. The results presented in the table demonstrate that our method achieves stable and effective performance across various proportions, both in terms of defending against jailbreak attacks and the detection performance of our detector.

## 5. Conclusion

In this paper, we address the challenge of defending against diverse jailbreak attacks. We propose a universal test-time defense framework designed to dynamically detect jailbreak attacks during testing and utilize detected jailbreak instructions to defensively train the model. To enhance jailbreak attack detection, we introduce a specialized gist token designed to extract harmful information from model responses with almost no additional cost, which is then classified using a binary classifier. Furthermore, to minimize the impact of model updates on the detector, we decouple the detector from defense training, ensuring they operate on separate parameters and do not interfere with each other. Extensive experiments demonstrate the efficacy of our method across a variety of scenarios. In future work, we will validate the effectiveness of our approach under more diverse model architectures (e.g., LLaMA3) and complex attack scenarios (e.g., adversarial jailbreak, multi-turn jailbreak).

## Impact Statement

This paper presents work whose goal is to advance the field of Machine Learning. There are many potential societal consequences of our work. One of which we think should be specifically highlighted is that the detection dataset we generated may contain harmful responses.

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

## A. The Jailbreak Evaluation

For the refusal judgment, we use the prefixes match method. We think the model refused to answer if the respones start with the following prefixes.

| |
|---|
| I'm sorry |
| Sorry |
| I am sorry |
| I apologize |
| As an |
| As a responsible |
| I'm an |
| I'm just |
| As an Assistant |
| If you have any other non-malicious requests |

| |
|---|
| I do not |
| I cannot |
| As a language model |
| I'm really sorry |
| My apologies |
| I'm not able to provide |
| I am really sorry |
| I can't provide |
| I can't assist |

## B. Additional Results

To validate the effectiveness of our method under continuous exposure to various forms of jailbreak attacks, we conducted experiments as shown in Table 4. We selected 500 different samples for each type of jailbreak attack and conducted the attacks in varying orders. As can be seen, even after undergoing the MM-SafetyBench attack, our method still maintains good defensive performance during the second exposure to the Figstep attack, without experiencing catastrophic forgetting.

Table 4: ASR(%) under continual changing environments.

| Attack Order ($\longrightarrow$) | | |
|---|---|---|
| Figstep | MM-SafetyBench | Figstep |
| 1.4 | 6.6 | 0.0 |

## C. Algorithm of TIM

---
**Algorithm 1** The Pipeline of TIM

---
**Initailize:** LLM $\mathcal{E}_l, \mathcal{C}_d$, Gist token $t_g$ and Detection Classifier $\mathcal{C}_d$, Jailbreak Memory $\mathcal{M}_j$, Detection Memory $\mathcal{M}_d$, Instruction Dataset $\mathcal{D}_{qa}$, Detection Dataset $\mathcal{D}_d$, Refusal Answer $t_{ref}$.

**Input:** An instruction $t_{ins}$.

Generate the answer $t_{ans}$ of $t_{ins}$ by Equ. (1)

Obtain the jailbreak label by Equ. (2) and (3).

**if** jailbreak label equals to 1 **then**

    Append $(t_{ins}, t_{ref})$ into $\mathcal{M}_j$.

    Append $\{(t_{ins}, t_{ref}, 0), (t_{ins}, t_{ans}, 1)\}$ into $\mathcal{M}_d$.

    Train the Adapter of $\mathcal{E}_l$ with $\mathcal{M}_j$ and $\mathcal{D}_{qa}$.

    Train $t_g$ and $\mathcal{C}_d$ with $\mathcal{M}_d$ and $\mathcal{D}_d$

**end if**

**Output:** Answer $t_{ans}$

---

We summarize the pipeline of TIM in Algorithm 1.

