# OpenReview forum: "Test-Time Immunization: A Universal Defense Framework Against Jailbreaks for (Multimodal) Large Language Models"
_ICML.cc/2025/Conference — Submitted to ICML 2025_

### Official Review · Reviewer_G2NZ · 2025-03-12

**Overall Recommendation:** 3

**Summary:**

This paper focuses on the task of jailbreak detection, which is based on the concern of large language models's vulnerability against jailbreaking attack. The paper proposed a method that is universal against different types of attacks. The assumption held by this paper is that detection is easier to implement for direct defense. The proposed method is tested on large benchmark like MM-SafetyBench.

**Claims And Evidence:**

Yes. The paper uses experiment results on large benchmark to support the claim.

**Essential References Not Discussed:**

N/A

**Experimental Designs Or Analyses:**

The paper is tested on large benchmark like MM-SafetyBench, which is helpful for verifying the soundness of the proposed mechanism.

**Methods And Evaluation Criteria:**

The proposed method makes sense. The proposed detection-based method is helpful for building a universal defense strategy.

**Other Comments Or Suggestions:**

N/a

**Other Strengths And Weaknesses:**

N/A

**Questions For Authors:**

Please refer to comments left for previous discussions.

**Relation To Broader Scientific Literature:**

Desipte the performance reported by the authors, the current contribution is incremental because jailbreaking attacks have been heatedly studied by experts in this domain. The proposed method mainly improves the defense method in the original setting.

**Theoretical Claims:**

N/A. This paper does not include proofs.

---

> ### Author Rebuttal · Authors · 2025-04-01
>
> Thanks for your reviews.
>
> > Desipte the performance reported by the authors, the current contribution is incremental because jailbreaking attacks have been heatedly studied by experts in this domain. The proposed method mainly improves the defense method in the original setting.
>
> We first build a framework that adaptively defends jailbreak attacks in an online manner, instead of static defense, which differs from previous works a lot. Moreover, we build an efficient detector and a dataset to train it, making our defense more practical.
>
> We sincerely thank you for your kind reviews.

---

### Official Review · Reviewer_iXZS · 2025-03-12

**Overall Recommendation:** 2

**Summary:**

This paper proposes Test-Time Immunization, a defense framework against jailbreak attacks for LLMs and multimodal LLMs. Specifically, this method actively collects jailbreak instructions during model deployment, then continues to improve the defense performance during deployment. Extensive experiments demonstrated that the proposed method achieves state-of-the-art performance against jailbreak attacks while maintaining utility.

**Claims And Evidence:**

Yes, the design goal is achieved by experiments.

**Essential References Not Discussed:**

N/A.

**Experimental Designs Or Analyses:**

Yes, I checked the soundness and validity of the experimental designs and analyses. There are two potential issues:

First, the number of jailbreak attacks tested is limited (one for MLLM and one for LLM), raising questions about the method's effectiveness against more advanced jailbreak techniques.

Second, the ASR calculation method using prefix matching is known to lack accuracy. It is suggested to implement LLM-based ASR calculation methods to validate the results more reliably.

**Methods And Evaluation Criteria:**

Overall, the proposed method makes sense. However, I have a question related to the pipeline.

The success of the defense appears to be largely dependent on the initial dataset D_d. If the proposed method cannot detect a sophisticated jailbreak attack, then it might never be able to capture it. Conversely, if there is too much data in the initial dataset, the system might become overly sensitive to benign prompts. Additional ablation on the choice of the initial dataset D_d could help address this concern.N/A. There is no theoretical claim.

**Other Comments Or Suggestions:**

All the details in Section 4 appear to be compressed into a confined space, making it difficult to follow. I recommend restructuring this section to improve clarity and readability.

**Other Strengths And Weaknesses:**

Strength: The experiments and ablation analysis are extensive, demonstrating the method's strong performance.

**Questions For Authors:**

How does the pipeline respond when it encounters a previously unseen jailbreak attack?

**Relation To Broader Scientific Literature:**

Compared to exisiting literature, this paper proposes a new test-time defense framework, which adaptively defend against jailbreak attacks. This method can inspire community in developing safer language models.

**Theoretical Claims:**

N/A. There is no theoretical claim.

---

> ### Author Rebuttal · Authors · 2025-04-01
>
> Thanks for your kind reviews. We provide additional experimental results in the link https://anonymous.4open.science/r/ICML109sda-E0E4/tab_and_fig.pdf. We will address your concerns one by one.
>
> >  If the proposed method cannot detect a sophisticated jailbreak attack, then it might never be able to capture it. Conversely, if there is too much data in the initial dataset, the system might become overly sensitive to benign prompts. Additional ablation on the choice of the initial dataset D_d could help address this concern.
>
> We designed the D_d to detect harmful responses instead of the jailbreak instruction. Although the attacker's jailbreak attack may be complex, the answer is not particularly difficult to identify. Moreover, as long as a small number of samples can be identified, we can use these small samples to implement our defense effectively according to our experiments.
>
>
> > The number of jailbreak attacks tested is limited (one for MLLM and one for LLM), raising questions about the method's effectiveness against more advanced jailbreak techniques.
>
> We supplemented the results of the white-box attack, GCG, in Table 3 in the link for reference. TIM still shows its effectiveness on GCG. After that, we evaluate two jailbreaks for LLM (i.e., GCG, I-FSJ) and two for MLLM (i.e., Figstep, MM-SafetyBench).
>
> > The ASR calculation method using prefix matching is known to lack accuracy. It is suggested to implement LLM-based ASR calculation methods to validate the results more reliably.
>
> We report the ASR evaluated by LLM in Table 3. Indeed, prefix match is more accurate than LLM, according to our observation. For example, VLGuard refused to answer the jailbreak question successfully, but most samples were still judged to have generated harmful content under LLM evaluation. Moreover, the attack success rates of the other three methods are lower than the actual situation. Methods based on large model evaluation are subject to fluctuations in many factors, whether it is the large model used, the template, or the decoding way.
>
> > How does the pipeline respond when it encounters a previously unseen jailbreak attack?
>
> All of our experiments are conducted under the unseen jailbreak attack, the response pipeline can be seen in Algorithm 1 in the appendix. Or if you want to ask the static transferability, like Reviewer 77PY, you can refer to the rebuttal for him.
>
> > Rewriting of Section 4.
>
> Thanks for your advice. We will rewrite Section 4 in our manuscript to make it clearer and more readable.

---

### Official Review · Reviewer_77PY · 2025-03-13

**Overall Recommendation:** 3

**Summary:**

The authors propose a novel defense framework for mono- and multi-modal generative models. in response to test-time LLM classifier defenses. The core novel contribution is their development of a detector for harmful outputs which uses an optimised gist token inserted at the end of a input + output pair to summarise whether the model's output was harmful. They train a binary classifier to recognise harmful gist tokens, and inputs that caused harmful model completions are stored in memory for future safety finetuning rounds.

**Claims And Evidence:**

* The paper claims that its method is more efficient than employing another LLM as input / output classifier, or augmenting prompts in other ways. In fact, they claim that such measures are "time-consuming and impractical." I think they're correct that there are important latency and cost considerations, but the paper does not provide sufficient comparative evidence of how much SOTA/similarly well performing classifiers would be, compared to their method. This leaves open the question of how effective their method truly is, which seems to me to be a core claim of why their method is worth employing. I'd expect this method to be more efficient, but evidence of this seems important.
* I'm also confused about the paper's focus on the efficiency of detection methods, given a comparative lack of focus on the efficiency of the test-time training they suggest to actually defend models. The paper does not sufficiently address whether test-time training is more or less efficient than its alternatives - I would be interested in discussion of factors such as latency, compute requirements, perhaps number of tokens).
* I'll address their claims about successful classification and defense in the next section.

**Essential References Not Discussed:**

Overall, as a discussion of defense against static, single-turn jailbreaks I think the literature cited is sufficient.

I'm a little surprised that they don't make reference to papers such as Anthropic's Rapid Response paper (Peng et al 2024, "Rapid Response: Mitigating LLM Jailbreaks with a Few Examples"), which is the main recent paper I'm aware of that addresses similar concerns about measuring the efficiency of different test-time jailbreak defense methods. A key metric that this paper used to measure efficiency was how many jailbreak samples are needed for each method, which would have been helpful for me to see more detailed discussion of in this paper (the authors include some discussion of ASR-50, and the comments on the number of jailbreaks needed to be seen from figstep to learn to reject that type of attack).

**Experimental Designs Or Analyses:**

I'm a little uncertain of how much the authors verified the accuracy of their binary harmfulness classifier post-training: I don't see mention of e.g. manual verification, or testing on held-out examples from the training sets they use (AdvBench and MM-SafetyBench).

**Methods And Evaluation Criteria:**

* I think it's hard to evaluate the results in this paper without discussion of the capability retention of these models when re-finetuned on the jailbreak samples that their classifier discovers. The paper does not sufficiently address the likely increase in false positive refusals, both in what is classified as harmful and in the subsequent outputs of the re-finetuned model.
* Given that their method relies on access to model weights, it's hard to evaluate how successful this defense would be when defending larger SOTA models. The authors only evaluate jailbreaks against three models (LLaVA-v1.6-Vicuna7B and LLaVA-v1.6-Mistral-7B), and LLaMA2-7B-chat. These are large and capable open-source models, but I don't think there is enough evidence that this is a practical defense method for frontier settings from just defending on these three models (especially only one text model). This is a challenge for all papers that work on open-weight models, but to be strongly convinced of the results in this paper I would at least need to see results on a wider array of models (and perhaps model sizes - to see if there might be trends or differences at different scales).
* Even if the authors couldn't have defended e.g. GPT-4 with their method, I would have been curious to hear how well their finetuning defense dataset could transfer.

**Other Comments Or Suggestions:**

I'm not sure the biological immunisation framing was very helpful for me to understand the paper, it took me a while to actually understand your method, and you could use this immunisation analogy even for classifier defenses that don't rely on gist tokens and internal states (which seems to me to be the main novel contribution here).

**Other Strengths And Weaknesses:**

The gist token approach for detecting harmful content is a novel contribution that seems to offer efficiency benefits compared to using separate classifier models.

The approach works on both text-only and multimodal models, which helps me to feel optimistic about transfer between very different kinds of model.

Their method seems to learn effective defenses after seeing relatively few examples (as shown by their ASR-50 results).

**Questions For Authors:**

1. Did you perform any manual verification of your classifier's outputs? I'm curious about whether you analyzed false positives/negatives beyond just reporting accuracy metrics.
2. I'd be curious about how well your fine-tuned models perform on new jailbreaks (do they get more robust even to held-out jailbreaks when finetuned against e.g. FigStep?)
3. How practical do you think your test-time training approach would be on frontier models? I'd be interested in understanding how you think your method would affect inference latency and training costs compared to alternatives, if you'd like to make claims about efficiency.
4. Do you have any evidence or theoretical basis for how your approach might perform on much larger models (e.g., 70B+ parameters)?
5. Have you measured how your fine-tuning process affects the model's ability to handle legitimate requests? Specifically, do you observe an increase in false positive refusals after adaptation?

**Relation To Broader Scientific Literature:**

The result of defending successfully (and classifying harmful outputs efficiently) is a reasonably compelling first result for a method that I'd need to see applying to more realistic jailbreak scenarios. The core idea of leveraging gist tokens to more efficiently train harmfulness classifiers seems like a promising direction for jailbreak defense systems to attempt to implement.

They mention it in the conclusion briefly, but I would be interested in more discussion about the realism of this defense method against more complex (and in my opinion more harmful jailbreaking methods). I'd appreciate, for example, discussion of how this method performs against multi-turn jailbreaks (e.g. PAIR, TAP, MSJ), or or what their defense does to change the scaling laws of the Best-of-N Jailbreaking paper from Anthropic (though I recognise that the last of these might not have been released when this paper was written).

**Theoretical Claims:**

No

---

> ### Author Rebuttal · Authors · 2025-04-01
>
> We are grateful for your valuable suggestions.  The figures and tables of the additional results are provided in the link https://anonymous.4open.science/r/ICML109sda-E0E4/tab_and_fig.pdf. We will do our best to address your concerns.
>
> > The concern about the efficiency and effectiveness of our detector. And the training cost of our defense training.
>
> To demonstrate the efficiency of our methods, we report the time cost in Table 1 (see the link). TIM takes a short time for detection (about 0.3% of inference cost), and the latency caused by detection is extremely low.
> For effectiveness, we compare metrics such as Accuracy, True Positive Rate (TPR), and False Positive Rate (FPR) with detection methods like Self-Defense and LVLM-LP in Table 3 of the original paper. Additionally, we report the detection metrics in Figures 3, 4, and 5b of the original paper. Our method shows effective detection performance (90+% TPR, and FPR less than 1%) compared to other methods and our variants. We also add the accumulated TPR and FPR during training in Figures 1 and 2 (in the link) for further analysis.
> According to the results in Table 1, the training time accounts for 12.2% of the total time used. The testing process of TIM is more efficient than the vanilla model because fine-tuned model can generate short rejection instead of long harmful responses for jailbreak instructions. For computation requirements, we only use one RTX A6000 for training. In practical applications, defense training can be conducted on the training GPU instead of the inference device.
>
> > The paper does not sufficiently address the likely increase in false positive refusals, both in what is classified as harmful and in the subsequent outputs of the re-finetuned model.
>
> We report the​ Over Defense Rate (ODR) to assess false positive refusals. Our method exhibits relatively few false rejections compared to other methods, as shown in our paper. In the experiment results, there are usually two types of samples that are judged as harmful: malicious instructions with harmful responses or rejection responses. Normal samples are rarely judged as harmful. Based on the ODR of our experiment, most samples can still be answered normally by our fine-tuned model.
>
> >Manual verification and testing on held-out examples.
>
> We think the detector performance can be validated by Acc, TPR, and FPR. For manual verification, the false positive examples are more like to be the samples metioned above and normal samples are rarely misjudged as jailbreak samples (see FPR of the TIM-NA). Moreover, we including the held-out accuracy of the validation set from detector training (99+% for all experiments) in the Table 6 (in the link).
>
> > Our method against more complex jailbreaks.
>
> The jailbreak method (i.e. PAIR, TAP) performs poorly (<10% ASR) on llama7b-chat, according to original paper and [1]. MSJ is a similar attack to the I-FSJ (which we used). Indeed, I-FSJ is a complex method including techniques like in-context jailbreak, greedy searching, and token replacement. I hope the explanation can address your concerns. Furthermore, we supplemented the results of the white-box attack, GCG in Table 3.
> [1] https://www.harmbench.org/results
>
> > How well your fine-tuned models perform on new jailbreaks (do they get more robust even to held-out jailbreaks when fine-tuned against?).
>
> It's worth noting that our method is an online adaptive defense method. New types of jailbreaks will be adaptively defended against as they emerge. Nevertheless, we demonstrate the static transferability of the fine-tuned model in Table 5 in the link. It is effective when migrating from a more complex attack (MM) to a simpler one (Figstep), but its effectiveness is limited in the reverse direction.
>
> > How practical do you think your test-time training approach would be on frontier models.
>
> The inference latency is extremely low, as stated in the first response. The training cost is only 12.2% of the total computing cost. Training occurs only when jailbreak activities are successful. Moreover, the defense training can be submitted as a background task. As a result, the latency is just the detection latency.
>
> > Do you have any evidence or theoretical basis for how your approach might perform on much larger models?
>
> We have conducted additional experiments using LLaVA1.6-Vicuna-13B, as presented in Table 4. TIM remains effective on the 13B model. However, due to computational resource limitations, we are unable to apply our method to 70B+ models (we only have access to an RTX A6000).
>
> > How fine-tuning process affect the model's ability to handle legitimate requests? Do you observe an increase in false positive refusals?
>
> We report ODR in our paper. TIM shows the fewest false positive refusals compared to other methods. For more details, you can refer to the change in ODR shown in Figures 1 and 2 in the link. There are increases, but acceptable.
>
> Morevoer, We will discuss Anthropic's paper in the manuscript.

---

> > ### Comment · Reviewer_77PY · 2025-04-03
> >
> > I thank the authors for their detailed rebuttal, and in light of the new experiments presented, I'm updating my score to a weak accept. I think my main update here comes from the new table 1, and from becoming more confident in the reported false positive rate.

---

> > > ### Author Response · Authors · 2025-04-07
> > >
> > > We are sincerely grateful for your recognition of our rebuttal and for raising your rating. We have further supplied the experiments conducted on the modern architecture LLaMA3-8B-Instruct against I-FSJ, which are shown here. TIM's ASR is reported as ASR / ASR-50. The results demonstrate that TIM is also effective while adopting LLaMA3 as the backbone.
> > >
> > > | Model | ASR | ODR |
> > > | -------- | -------- | -------- |
> > > |  Vanilla    |  94.3    | 0.2     |
> > > | TIM | 1.0/0.0  | 0.2 |
> > >
> > >
> > >
> > > ***
> > >
> > > *As the other reviewers have not yet commented on our rebuttal responses, we apologize for taking up this space to provide an additional response to them.*
> > >
> > > + @reviewer ydQk
> > >
> > > We have provided **the results you mentioned about modern architectures such as LLaMA3**, in the table above. The results demonstrate that TIM is still effective for LLaMA3-8B-Instruct. I hope this further enhances your recognition and confidence in the generalization of our work.
> > >
> > > Furthermore, we think your major concerns lie in 1) **the scalability of the gist token** and 2) **the potential degradation of TIM**.
> > > We have provided the avg. token length in training and test (296 for training and 1720 for test) to show that **gist token can extend to longer context**. Besides, we provide the performance curve during testing to demonstrate that **our method (TIM) doesn't suffer from performance degradation**.
> > >
> > > + @reviewer iXZS
> > >
> > > We think your main concern lies in 1) the evaluation metrics and 2) the number of jailbreak attacks.
> > >
> > > **We have provided additional experiments on the GCG attack and reported the ASR evaluated by LLM**, and we are looking forward to receiving your additional comments on our response.
> > >
> > > ***
> > > Once again, we would like to thank the AC and each reviewer for their efforts in the review process of this work.

---

### Official Review · Reviewer_ydQk · 2025-03-13

**Overall Recommendation:** 3

**Summary:**

In a nutshell, this paper introduces Test-Time Immunization (TIM) as a universal defense against jailbreak attacks on large language models. Specifically, the authors insert a special "gist token" which is used for binary classification of spotting harmful outputs, i.e., question, answer, gist token to predict the harmful output. Once an attack is detected, the model is fine-tuned online with safe refusal responses by using LoRA to preserve its regular performance (i.e., prevent overfit).

**Claims And Evidence:**

The overall claim is clear and sounds.

**Essential References Not Discussed:**

I think it is not a weakness (it is more like a suggestion) but it is great to discuss the difference between recent test-time defense methods [1,2] in the final revision. Especially [1] shares a very similar idea in my perspective.

[1] Backtracking improves generation safety, ICLR 2025\
[2] Trading Inference-Time Compute for Adversarial Robustness, arXiv 2025

**Experimental Designs Or Analyses:**

I have checked the experimental designs and analysis, which looks reasonable and clear.

I think it would be great to add some modern architectures (e.g., Llama3, Gemma2, Qwen2.5) rather than Llama2. But I don't think this is a major issue.

**Methods And Evaluation Criteria:**

One of the major beliefs of adversarial robustness evaluation is to attack the defense system itself [1,2,3]. However, from my understanding, this paper did not consider 'adaptive attack', i.e., attacking the proposed method.

[1] Obfuscated gradients give a false sense of security: Circumventing defenses to adversarial examples, ICML 2018
[2] On Evaluating Adversarial Robustness, arXiv 2019
[3] On adaptive attacks to adversarial example defenses, NeurIPS 2020

**Other Comments Or Suggestions:**

I think this paper is slightly in the acceptance side. It would be great if the authors can address some concerns and questions during the rebuttal.

**Other Strengths And Weaknesses:**

**Strengths**

The overall paper is well-written except for the mathematical formulation (see below).

The overall method sounds and using gist token makes sense.

---

**Weakness**

This question can be related to my misunderstanding (also see the question part regarding adapter training). I wonder how much the model degrades over time, i.e., multiple attack detections. Since the model is trained for every detected jailbreaking attack, I am concerned about catastrophic forgetting. In the rebuttal, it would be great if the authors can report the performance change throughout the fine-tuning. I do agree that LoRA prevents some part, but still I think this part should be highlighted.

One concern is the scalability of gist tokens in multi-tern scenarios or generalizations (e.g., long contexts). Specifically, we need to insert the gist token every time to detect jailbreaking. While this paper primarily focuses on single turn scenario, it is somewhat concerned with how to train or how to insert the gist token for multi-turn cases (e.g., insert to every user query?). Also, one question will be, do this gist token work for longer context size than it is originally trained?

I think the paper could write a better mathematical formulation for test-time training (section 4.3). Since there is no objective or mathematical formulation, it is somewhat hard to understand. I had to read the Algorithm table in the Appendix to fully understand the method. In the revision (or in the final revision), I kindly ask the authors to re-write this part.

**Questions For Authors:**

When training with detection loss function in Equation 4, do the authors (i) only train the gist token or (ii) the full network?

Do we need to train the adaptor from scratch every time when the attack is detected, or is it reused (continuously trained?) If the LoRA is trained from scratch every time, I have some concern regarding training cost (as the memory bank grows).

**Relation To Broader Scientific Literature:**

The paper tackles an interesting direction by test-time fine-tuning the model to prevent jailbreaking attacks online.

**Theoretical Claims:**

The paper does not provide a theoretical claim (I don't think this theory is necessary for this case).

---

> ### Author Rebuttal · Authors · 2025-03-31
>
> We sincerely appreciate the constructive feedback. Below we provide point-by-point responses with methodological clarifications and supplementary experimental evidence. All referenced figures/tables are available in our link https://anonymous.4open.science/r/ICML109sda-E0E4/tab_and_fig.pdf.
>
> > Regarding adaptive attack against our method.
>
> We acknowledge the reviewer's concern about possible adaptive attacks targeting our defense approach. Indeed, adversaries could attempt to craft adversarial samples to mislead our detector and degrade its performance. However, in practical deployment scenarios, our detector operates as a black box—its internal decisions are not directly exposed to users. This inherent opacity increases the difficulty of mounting successful attacks, as adversaries lack direct feedback to refine their adversarial inputs. However, we are considering designing a targeted attack in future work to verify whether our method is robust enough.
>
> > Concerned about performance during finetuning and the catastrophic forgetting.
>
> In Figures 1 and 2 in the provided link, we report key metrics, including accumulated ASR, ODR (Over Defense Rate), TPR, and FPR, for our method during the fine-tuning process. If the reviewer is referring to the model's ability to answer normal questions, the changes in ODR provide a clear indication of its performance in this regard. While our method did cause the model to reject some normal answers during testing, this effect was manageable and did not lead to significant degradation in overall performance.
> If you mean the ability to reject the jailbreak attack encountered before, we reported this result in Figure 4 of the original paper. Despite experiencing other attacks, we still maintained our defense against the previous jailbreak attack. No obvious catastrophic forgetting of defense capability against previous jailbreaks is observed.
>
> > Scalability of gist token.
>
> For multi-turn scenarios, the purpose of gist token is to detect whether the model generates malicious answers, so in multi-turn conversations, we only need to insert gist token after each answer generated by the model, and remove it from the KV cache after the detection is completed. For detector training, we can also build a multi-turn conversation detection dataset to ensure that we can still maintain effective detection when facing multi-turn data.
> For generalizations for longer context, in our experiments, I-FSJ is a in-context jailbreak attack that uses multiple demonstration to induce our model to generate malicious answers. The average length of its jailbreak instructions is 1720 tokens, while in our training set, the average length of questions and answers is only 15.2 and 281.4 tokens. However, we can still effectively detect jailbreak samples in I-FSJ, hoping that this can address your concern about the generalization of gist token.
>
> > Training parameter of the detection loss.
>
> We solely train the gist token and the binary classifier. During the training of the detector, the LLM is frozen to prevent any impairment to the model's question-answering ability.
>
> > Question about the defense training.
>
> The LoRA is continuously trained. We have attached the training and detection costs in Table 1, which can be found via the provided link. The results indicate that our method is highly efficient and thus practical for application.
>
> > Modern architecture.
>
> We have added the results of LLaVA-v1.6-Vicuna-13B in Table 3, which can be found in the link. LLaVA 1.6 is among the most state-of-the-art MLLMs. Currently, we are attempting to conduct experiments on LLaMA3-8B-Instruct, and we will update the results soon if we finish.
>
> > Related works and writing.
>
> Thank you for your valuable advice. In the final version, we will carefully consider and discuss the differences between the works you have mentioned. Additionally, we will rewrite Section 4.3 to enhance its clarity and comprehensibility.
>
> We sincerely hope that the above rebuttal effectively addresses your concerns.
>
> If you have additional question or we midunderstand your review, feel free to respond.

---

### Decision · Program_Chairs · 2025-05-01

**Decision:**

Reject

**Comment:**

The authors propose a novel defense framework for mono- and multi-modal generative models. in response to test-time LLM classifier defenses. The core novel contribution is their development of a detector for harmful outputs which uses an optimised gist token inserted at the end of a input and output pair to summarise whether the model's output was harmful. They train a binary classifier to recognise harmful gist tokens, and inputs that caused harmful model completions are stored in memory for future safety finetuning rounds.

However, there are several points to be further improved. For example, there are concerns about the scalability of gist tokens in multi-tern scenarios or generalizations (e.g., long contexts). Specifically, we need to insert the gist token every time to detect jailbreaking. The paper could write a better mathematical formulation for test-time training (section 4.3). Since there is no objective or mathematical formulation, it is somewhat hard to understand. The success of the defense appears to be largely dependent on the initial dataset D_d. If the proposed method cannot detect a sophisticated jailbreak attack, then it might never be able to capture it. Conversely, if there is too much data in the initial dataset, the system might become overly sensitive to benign prompts. It's hard to evaluate the results in this paper without discussion of the capability retention of these models when re-finetuned on the jailbreak samples that their classifier discovers. The paper does not sufficiently address the likely increase in false positive refusals, both in what is classified as harmful and in the subsequent outputs of the re-finetuned model.

Therefore, this paper cannot be accepted at ICML this time, but the enhanced version is highly encouraged to submit other top-tier venues.